# *Kcnj16* (Kir5.1) Gene Ablation Causes Subfertility and Increases the Prevalence of Morphologically Abnormal Spermatozoa

**DOI:** 10.3390/ijms22115972

**Published:** 2021-06-01

**Authors:** Giulia Poli, Sonia Hasan, Silvia Belia, Marta Cenciarini, Stephen J. Tucker, Paola Imbrici, Safa Shehab, Mauro Pessia, Stefano Brancorsini, Maria Cristina D’Adamo

**Affiliations:** 1Section of Pathology, Department of Medicine and Surgery, University of Perugia, 06132 Perugia, Italy; poligiulia.mail@gmail.com (G.P.); stefano.brancorsini@unipg.it (S.B.); 2Department of Physiology, Faculty of Medicine, Kuwait University, Safat 13110, Kuwait; sonia.hasan@ku.edu.kw; 3Department of Chemistry Biology and Biotechnology, University of Perugia, 06123 Perugia, Italy; silvia.belia@unipg.it; 4Section of Physiology & Biochemistry, Department of Medicine and Surgery, University of Perugia, 06132 Perugia, Italy; marta.cenciarini@live.it; 5Clarendon Laboratory, Department of Physics, University of Oxford, Oxford OX1 3PU, UK; stephen.tucker@physics.ox.ac.uk; 6Department of Pharmacy-Drug Sciences, University of Bari ‘‘Aldo Moro”, 70125 Bari, Italy; paola.imbrici@uniba.it; 7Department of Anatomy, College of Medicine and Health Sciences, United Arab Emirates University, Al Ain P.O. Box 17666, United Arab Emirates; s.shehab@uaeu.ac.ae; 8Department of Physiology & Biochemistry, Faculty of Medicine and Surgery, University of Malta, MSD 2080 Msida, Malta; mauro.pessia@um.edu.mt; 9Department of Physiology, College of Medicine and Health Sciences, United Arab Emirates University, Al Ain P.O. Box 17666, United Arab Emirates

**Keywords:** Kir4.1, *KCNJ10*, Kir4.2, *KCNJ15*, *KCNJ16*, Kir5.1, potassium channel, testis, epididymis, spermatozoa, flagellar morphology, sperm motility, male fertility

## Abstract

The ability of spermatozoa to swim towards an oocyte and fertilize it depends on precise K^+^ permeability changes. Kir5.1 is an inwardly-rectifying potassium (Kir) channel with high sensitivity to intracellular H^+^ (pHi) and extracellular K^+^ concentration [K^+^]_o_, and hence provides a link between pHi and [K^+^]_o_ changes and membrane potential. The intrinsic pHi sensitivity of Kir5.1 suggests a possible role for this channel in the pHi-dependent processes that take place during fertilization. However, despite the localization of Kir5.1 in murine spermatozoa, and its increased expression with age and sexual maturity, the role of the channel in sperm morphology, maturity, motility, and fertility is unknown. Here, we confirmed the presence of Kir5.1 in spermatozoa and showed strong expression of Kir4.1 channels in smooth muscle and epithelial cells lining the epididymal ducts. In contrast, Kir4.2 expression was not detected in testes. To examine the possible role of Kir5.1 in sperm physiology, we bred mice with a deletion of the *Kcnj16* (Kir5.1) gene and observed that 20% of Kir5.1 knock-out male mice were infertile. Furthermore, 50% of knock-out mice older than 3 months were unable to breed. By contrast, 100% of wild-type (WT) mice were fertile. The genetic inactivation of *Kcnj16* also resulted in smaller testes and a greater percentage of sperm with folded flagellum compared to WT littermates. Nevertheless, the abnormal sperm from mutant animals displayed increased progressive motility. Thus, ablation of the *Kcnj16* gene identifies Kir5.1 channel as an important element contributing to testis development, sperm flagellar morphology, motility, and fertility. These findings are potentially relevant to the understanding of the complex pHi- and [K^+^]_o_-dependent interplay between different sperm ion channels, and provide insight into their role in fertilization and infertility.

## 1. Introduction

Ion channels are important for maintaining the electrochemical gradient across the cell membrane. In particular, the inwardly-rectifying potassium (Kir) channels provide K^+^ fluxes that play crucial roles in almost all kinds of cells, including the maintenance of the cell membrane potential (Vm) and cellular excitability [1]. To date, there are seven Kir subfamilies (Kir1.0–Kir7.0) and at least fifteen different Kir genes within these subfamilies [1,2,3]. The functional roles of most of these Kir subfamilies have already been established. Nevertheless, attempts to identify and decipher the functional role of some Kir channel types have been hampered by the lack of highly specific blockers. Genetically modified animals, in which the expression of distinct proteins can be controlled, would therefore represent suitable models to explore the physiological role of the relevant gene product. By using brain slices dissected from *Kcnj16* knock-out (KO) mice, we demonstrated that the Kir5.1 channel plays a crucial role in linking the excitability of brain *locus coeruleus* neurons to changes in pHi [4]. In fact, there is an emerging consensus on the role that Kir5.1 plays in the regulation of pH-dependent cellular processes, and in the control of K^+^ fluxes during acid–base changes. Notably, Kir5.1 confers several unique functional properties when it co-assembles selectively with either Kir4.1 or Kir4.2 subunits to form novel heteromeric channels [5,6,7,8]. Most importantly, the heteromeric assembly dramatically increases channel sensitivity to regulation by intracellular H^+^ (pHi), but not to extracellular pH [5,6,7,8]. While many Kir channels also exhibit some degree of pH sensitivity, heteromeric Kir4.1/Kir5.1 and Kir4.2/Kir5.1 channels are exceedingly sensitive to pHi within the physiological range [7] thereby providing a mechanistic link between pHi changes and membrane potential. Furthermore, in agreement with the Nernst equation, changes in the extracellular K^+^ concentration [K^+^]_o_ shift the reversal potential of these Kir5.1-dependent currents [2,5,6,7].

A mounting body of evidence indicates that ion channels regulate the maturation and motility of spermatozoa, and actively take part in the physiological processes necessary for fertilization, such as capacitation and the acrosome reaction, whereby acrosomal hydrolytic enzyme release allows for the digestion of the oocyte zona pellucida. Several ion channel types such as voltage-dependent Ca^2+^, Na^+^, K^+^, and H^+^ channels have been cloned and described in testes, where they are involved in spermatogenesis. In spermatozoa, they also play important roles in processes such as motility, capacitation, and acrosome reaction. [9,10,11]. Kir5.1 mRNA distribution analysis by RT-PCR indicates this subunit is expressed in rodent testes [2]. Immunoreactivity studies allowed for the localization of Kir5.1 in rat seminiferous tubules, in spermatogonia, both primary and secondary spermatocytes, spermatids, and in the spermatozoan head and tail [12]. The intensity of Kir5.1 immunofluorescence increases with age at every stage in the development of sperm from spermatogonia. Overall, the evidence demonstrated that Kir5.1 expression in the testis is localized to cells involved in spermatogenesis, sperm development, and maturation such that expression is optimal during mammalian sexual maturity [12]. The importance of Kir5.1 and the role it plays in sperm morphology, maturity, motility, and fertility is, however, unresolved.

To investigate the contribution of Kir5.1 to fertility and morphology of mammalian spermatozoa, we bred mutant Kir5.1 KO mice with the specific deletion of *Kcnj16* and collected sperm from the epididymis. Here we report that genetic inactivation of *Kcnj16* results in smaller testes, a greater proportion of sperm with folded flagellum and subfertility compared to WT littermates. Unexpectedly, the abnormal sperm that retained motility displayed increased progressive motility. These findings identify Kir5.1 as a key contributing element for spermatozoa with normal flagellar morphology and motility that could be potentially relevant for the study and diagnosis of human fertility.

## 2. Results

### 2.1. Expression of Kir4.1, Kir4.2, and Kir5.1 Channels in Epididymal Ducts

Cauda epididymis was collected from 2-month-old C57BL/6J WT mice and immunoreactivity was determined using Anti-Kir4.1, -Kir4.2, and -Kir5.1 antibodies. Immunocytochemistry results indicated strong expression of Kir4.1 in smooth muscle and in epithelial cells lining the epididymal ducts (Figure 1A,B). By contrast, no Kir4.1 immuno-reactivity was observed in epididymal sperm (Figure 1). The absence of Kir4.1 immunoreactivity was not due to lack of spermatozoa, which were present in high number in the epididymis (Appendix A). By using anti-Kir4.2 antibodies, we were unable to detect any expression of Kir4.2 channels in mouse testes (not shown). On the other hand, strong Kir5.1 immunoreactivity was detected in spermatozoa located in the cauda epididymis (Figure 1C,D), confirming previous findings [12].

### 2.2. Ablation of Kir5.1 Gene Affects Fertility and Testicular Development

The breeding performance of *Kcnj16* KO mice was assessed during a 2 year period. Overall, we observed that 20% of mutant male mice were infertile. Furthermore, 50% of KO animals older than 3 months were unable to breed. By contrast, 100% of WT mice were fertile, only becoming infertile after 24 months of age, consistent with previous findings [13]. No statistical differences in the litter size were observed between the nests sired by mutant and WT male mice. Remarkably, the genetic inactivation of *Kcnj16* also resulted in smaller testes. Indeed, the weight and volumes of testes collected from 2-month-old KO mice were significantly smaller than that of the WT (Figure 2A,B). The body weight of WT and mutant mice, from which the testes were collected, was not statistically different (26.8 ± 0.8 g vs. 26.0 ± 1.7 g; *p* > 0.05). These results excluded the possibility that the mutation could affect the weight and volume of the testis by reducing the overall body weight of the animals.

Next, we tested the hypothesis as to whether Kir5.1 channels could contribute to establishing the correct pH of the fluid present in the epididymis. However, when the luminal contents of cauda epididymal fluid were immediately spread onto pH strips, the pH values obtained for the mutant and WT mice were not statistically different (6.6 ± 0.1 vs. 6.7 ± 0.1; *n* = 8; *p* > 0.5). We also examined sperm viability and progressive motility, namely active cellular movement, either linearly or in a large circle, regardless of speed. Figure 3 depicts the viability and progressive motility of sperm collected from Kir5.1 KO and WT mice. Genetic ablation of *Kcnj16* did not statistically affect viability and progressive motility (*p* > 0.05; Figure 3A,B).

### 2.3. Kir5.1 KO Mice Showed Increased Percentage of Sperm with Abnormal Flagellar Morphology

To determine possible morphologic abnormalities of sperm caused by the absence of Kir5.1 channels, spermatozoa were collected from WT and KO mice by placing minced cauda epididymis in a modified Krebs–Ringer medium. WT and mutant sperm shape (head, mid-piece, and tail morphology) are represented in Figure 4A,B, respectively (1000 sperm cells, *n* = 5).

Notably, we observed that the percentage of spermatozoa that displayed normal morphology was decreased in Kir5.1 KO (39.6%) mice compared to WT (54.1%) (*p* < 0.01). Consistently, the percentage of spermatozoa that displayed folded flagellar shape was increased in samples collected from Kir5.1 KO mice (60.4%) compared to WT (45.9%) (Figure 5; *p* value < 0.01).

### 2.4. Spermatozoa from Kir5.1 KO Mice with Altered Flagellar Morphology Show Increased Motility

The evidence that some Kir5.1 KO mice were either infertile or subfertile prompted us to investigate altered motility in the sperm that had abnormal flagellar morphology. We observed that spermatozoa with folded flagella were motile. Unexpectedly, the percentage of sperm from Kir5.1 KO mice with abnormal morphology displayed progressive motility significantly higher (*p* < 0.01) than sperm with folded tails from WT (Figure 6). 

## 3. Discussion

The capacity of spermatozoa to swim towards and fertilize an oocyte requires precise changes in K^+^ permeability across the cell membrane, and Kir5.1 channels appear to play a key role in these processes. In particular, the fact that this flux is dependent on pHi, a vital cue and key regulator of these K^+^ fluxes, strongly implicates Kir5.1. In the present study, we examined the role of the Kir5.1 channel in the reproductive performance of rodents, as well as in their sperm morphology and motility. Our results showed that genetic deletion of Kir5.1 resulted in reduced fertility, smaller testes, and increased aberrant flagellar morphology. Surprisingly, there was also a significant enhancement in the progressive motility of spermatozoa with abnormal morphology.

The presence and specific localization of many different ion channel types in sperm indicate the important role they may play in the sophisticated physiology of sperm. Kir5.1 immunoreactivity was observed in spermatozoa from the caput epididymis. Surprisingly, both anti-Kir4.1 and anti-Kir4.2 antibodies failed to detect their respective subunits in spermatozoa, suggesting that Kir5.1 may not form heteromeric Kir4.1/Kir5.1 and Kir4.2/Kir5.1 channels in murine sperm as it does in most other cell types [5,7]. However, mRNA for Kir4.1 and Kir4.2 subunits have been found in human spermatozoa [14,15,16,17]. Based on these findings, it is therefore likely that Kir4.1/Kir5.1 and Kir4.2/Kir5.1 channels are assembled in human spermatozoa, although direct evidence showing the expression of these channel types in human sperm is, to the best of our knowledge, not currently available. Intriguingly, strong Kir4.1 expression was detected in epithelial cells lining the epididymal ducts and peritubular smooth muscle in the cauda epididymis in mice. These observations suggest that Kir4.1 could potentially play an important physiological role in these tissues.

The deletion of the *Kcnj16* totally abolished fertility in a significant population of male mice. Furthermore, with increasing age, the fertility of the remaining KO mice declined dramatically and was consistent with the abnormal testicular development and anomalous sperm morphology we observed. This overt phenotype of Kir5.1 KO mice suggests that Kir5.1 channels are not dispensable and cannot be compensated by other members of the Kir family, or any other K^+^ channel, and this may be related to their extreme sensitivity to intracellular pH [5,7,18]. Cytoplasmic acidifications strongly inhibit these channels, leading to membrane depolarization, whereas alkalization would hyperpolarize the membrane potential [7,18]. However, changes in extracellular pH do not affect this particular channel type. Consistent with this, a K^+^-selective inwardly-rectifying current was previously isolated in mouse spermatogenic cells and, interestingly, a cytosolic acidification reversibly inhibited this current [9].

Mammalian epididymal spermatozoa are quiescent with an acidic internal environment (pH_i_~6.8) [19]. In these circumstances, Kir5.1 channels would be inhibited consistent with the relatively depolarized potential found in functionally immature spermatozoa. During capacitation, pHi alkalization would increase Kir5.1 channel activity and hyperpolarize the membrane. It is noteworthy that Ba^2+^, a Kir channels blocker, impedes hyperpolarization during capacitation [9,20]. Additionally, it is during capacitation that the spermatozoa display enhanced motility, whereby the angle of flagellar bend increases, resulting in the whip-like motion underlying its swimming force. It is during this process that spermatozoa bind to the zona pellucida of the oocyte, and where additional increases in sperm pH_i_ result in the acrosome reaction which allows oocyte penetration [19,21,22,23,24].

It has been shown that in vitro capacitation of human spermatozoa occurs at the same time as an increase in H^+^ currents through Hv1 channels (*HVCN1*) located in the flagellum [25]. Proton extrusion via Hv1 channels would produce the intracellular alkalization required for Kir5.1 channel opening and hyperpolarization-induced flagellar rotation. Interestingly, flagellar membranes also possess CatSper (a voltage-dependent calcium channel), that is also activated at alkaline pH. Being voltage-dependent, both CatSper and Hv1 channel activity would also be highly sensitive to changes in Vm caused by Kir5.1. Notably, CatSper and Hv1 ensure fast signal transduction along the length of the whole flagellum and are critical for hyperactivation, rheotaxis, and the rolling motion that enables sperm to swim in a straight trajectory despite its biased waveform [26]. It is therefore possible that crosstalk between Hv1, Kir5.1, and CatSper channels may fine-tune each other to control spermatozoa shape and motility, and the absence of Kir5.1 would alter this balance. However, the molecular mechanisms that control sperm pHi in mice and humans appear to be different; mouse spermatozoa express much lower levels of Hv1 channels compared to human [25], and Hv1 knock-out mice do not display any fertility phenotype [27]. Thus, the identity of the protein responsible for pHi alkalization in mouse sperm, and the physiological role of these H^+^ currents, remains elusive. It has been proposed that the pH regulation of Vm is mediated by the SLO3/LRRC52 complex that underlies the mouse Ca^2+^-activated K^+^ current I_Ksper_. However, in humans, I_KSper_ activation is independent of intracellular alkalization [19] but rather strongly activated by Ca^2+^ [28], thus supporting the potential role for Kir5.1 in linking Vm to pHi.

Extracellular environmental cues are also important for sperm during fertilization. Human tubal fluid contains 4–5-fold higher K^+^ concentrations than serum (serum: 3.6–5 mM; fallopian tube: 23–30 mM) [29] and this would also influence Vm through its effect on K^+^ channels. Thus, Kir5.1 may provide a further link between changes in the extracellular K^+^ to Vm. This mechanism could also account for the slight increase in motility that was observed in morphologically normal KO sperm. Possible compensatory mechanisms developed by a subpopulation of the latter KO sperm could account for the insignificant difference of their progressive motility (Figure 3B).

It is noteworthy that a significant increase in spermatozoa with abnormal tails, with strikingly similar morphologies to those reported in this study, were also observed in mice lacking Dystrophin Dp71. The absence of Dp71 results in α-syntrophin relocalization in spermatozoa, and disruption in the distribution of the Kir4.1 potassium channel in Müller glial cells [30,31]. It has been proposed that the Dp71-α-syntrophin complex may participate in the scaffold that contributes to anchor K^+^ and Na^+^ channels at the head and middle piece of spermatozoa. The altered distribution of ion channels and signaling proteins in Dp71 mutant mice has been put forward as the mechanism responsible for the production of aberrant flagella [32]. This suggests an alternative possibility whereby the absence of Kir5.1 could alter the distribution/expression/activity of proteins which interact with these scaffolds (e.g., Na^+^ channels), resulting in folded flagella, altered cell excitability, and enhanced motility.

It has been estimated that infertility affects ~10–13% of people of reproductive age [33] and sperm ion channel dysfunction has significant consequences for fertilization. Indeed, ~10% of patients undertaking in vitro fertilization (IVF) procedures or intracytoplasmic sperm injection had sperm with depolarized membrane potentials (*V*m ≥ 0 mV) that were associated with a low fertilization rate following IVF [34]. Asthenoteratozoospermia is a disorder characterized by absent, short, bent, coiled, or irregular flagella and results in seriously impaired sperm motility [35]. Genetic defects in CatSper1/2 [11,36] and in the sperm-specific anion exchanger SLC26A8 that associates with the CFTR channel [37,38], have been identified in patients affected by distinct types of asthenozoospermia. Additionally, slight changes in the mRNA levels for Kir4.2 channels have been observed in human sperm from teratozoospermic individuals with fertility problems compared to normospermic subjects [16,17].

In conclusion, male Kir5.1 KO mice display a clear reduction in fertility. The implication of Kir5.1 in spermatozoa morphology and motility, its expression in every stage of sperm development, and likely role in the complex signaling cascades involved in fertilization of an oocyte suggest *KCNJ16* as an attractive target for genetic screening of patients that exhibit abnormal testicular development, altered sperm morphology, and subfertility.

## 4. Materials and Methods

### 4.1. Creation of a Colony, Mice Genotyping, and Breeding of Kir5.1 (Kcnj16) KO Mice

A colony of C57BL/6J mice carrying the null allele *Kcnj16*^(−/−)^ was established by breeding heterozygous male and female mutant mice *Kcnj16*^(+/−)^ [4]. *Kcnj16*^(−/−)^ mice were backcrossed to the appropriate WT mice every ten generations to maintain genetic background. The mice were kept in polypropylene cages at room temperature (22 °C) with a normal 12 h light/dark cycle. They were given standard commercial food and water ad libitum. Mice were genotyped as previously described [4]. Briefly, DNA from tail biopsies was analyzed by PCR using a three-primer set. A 225-bp fragment from the WT Kir5.1 gene was amplified using a forward primer (5′-CTGCTTGCAGTTTGAAGGAAG-3′). This corresponds to codons 325–331 of the mouse Kir5.1 gene and a reverse primer (5′-CATTCATCTTGTGGGGACAGGACGGTCT-3′) corresponding to anticodons 389–397. A 325-bp PCR product from the successfully targeted gene was amplified using the reverse primer from the Kir5.1 gene and a forward primer (5′-AGGGGGAGGATTGGGAAGACAATAGCA-3′) complementary to sequences in the 3′ region of the integrated neomycin resistance gene. PCR cycle parameters were 94 °C for 30 s, then 30 cycles of 94 °C for 15 s, 60 °C for 20 s, and 72 °C for 40 s. Samples were run on a 1.8% agarose gel. The investigation and animal protocols were approved by the national and institutional ethics Committees (authorization number 226/2018-PR approved on 14 March 2018).

### 4.2. Fertility Assessment

To analyze the fertility of mice, we paired *Kcnj16*^(−/−)^ and *Kcnj16*^(+/+)^ male mice (2 months of age at the starting of the breeding procedure; *n* = 10 in each group), with *Kcnj16*^(+/+)^ female mice. All female mice were proved fertile before being taken into the experiment. If a female died, became ill, or became old, it was replaced. The ages of the females used ranged from 8 weeks to 7 months. Mice were housed as one male and one female per cage. Over a period of two years, we recorded the number of pregnancies and deliveries performing a correlation with the age of male breeders. We scored the males that failed in siring litters as sterile, the males in which litters sired declined after approximately 3–5 months as subfertile, and the males able to produce litters at the age up to 20 months as fertile. The numbers of pups were counted for each delivery.

### 4.3. Tissue Collection and Immunohistochemistry

C57BL/6J mice were deeply anaesthetised with sevoflurane, and testicles were collected from adult (P60 ± 10 days) *Kcnj16*^(−/−)^ and *Kcnj16*^(+/+)^ male mice. The testis and epididymis were dissected out and immediately fixed by immersion in 4% paraformaldehyde in phosphate buffer (PB, pH 7.4) overnight. The tissues were then stored in 30% sucrose in PB overnight. Cryostat sections (15 μm) were collected on gelled slides, rinsed in phosphate buffer saline (PBS, pH 7.4), and then incubated overnight in rabbit anti-Kir4.1, anti-Kir4.2, and anti-Kir5.1 antibodies (Alomone labs, Jerusalem, Israel, cat#: APC-035, APC-058, APC-123) diluted 1:1000 in PBS containing 0.3% Triton. After rinsing with PBS, the sections were incubated with anti-rabbit conjugated to Alexa 488 diluted 1:200 in PBS (Jackson Immuno Research, West Grove, PA, USA) for 1 h. Sections were mounted in antifade immuno-mount (Thermo Fisher Scientific, Runcom, Cheshire, UK) and examined with a Nikon fluorescent microscope equipped with appropriate filters and a Nikon C1 laser scanning confocal microscope (Nikon, Tokyo, Japan). Representative digital images were captured and the resulting files were used to generate figures in Adobe Photoshop software CS6 (San Jose, CA, USA) where photomicrographs were adjusted for contrast and brightness.

### 4.4. Measurement of Testicular Volume

Four adult WT and Kir5.1 KO mice, with an average age of 60 ± 10 days, were used. The animals were anesthetized via intramuscular injection of ketamine hydrochloride and xylazine cocktail (87.5 mg/kg Ketamine-12.5 mg/kg Xylazine) and testes collected and weighted using an analytical balance. Testicular dimensions were measured with an electronic caliper (Sattiyrch Digital Caliper, Goldmoon Industry, Suzhou, Jiangsu, China) and testicular volumes were calculated using the formula for the calculation of a solid ellipsoid triaxial body, as previously described: 4πabc/3 [39,40,41]. This formula was used considering π = 3.14; a = half the height; b = half the width; c = half the length.

### 4.5. Measurements of pH in Epididymal Fluids Using pH Strips

The pH of epididymal fluid was measured, as previously described [42]. To avoid any loss of carbon dioxide during collection, the pH of the fluid was determined immediately after release from the epididymal lumen. The mice were killed by cervical dislocation and the cauda epididymidis was isolated. Lobules of tubules were dissected free of the capsule to avoid blood vessels. Tubules were cut at a few sites and gently pressed, so that the exuded luminal contents were immediately smeared onto a pH strip and read (with 0.2 unit increments in the reference colours; Hydrion pH strips from Microfine, Micro Essential laboratory, New York, NY, USA).

### 4.6. Sperm Analysis

Animals were sacrificed by carbon dioxide overdose. Their left caudal epididymis was removed immediately by cutting at the intersection of the epididymis and vas deferens. In order to release the sperm from the ependymal tissue, the tissue was submerged in 1 mL of warmed Biggers–Whitten–Whittingham (BWW) medium, and was then incubated for 10 min at 37 °C. Epidydimal tissue was then removed, and the solution was gently swirled. A total of 20 µL of live sperm medium were placed in 100 µm-deep haemocytometer chambers. Improved Neubauer haemocytometer chamber was used for sperm count and progressive motility analysis by light microscopy (Nikon Eclipse E600; Nikon Corporation, Tokyo, Japan). Sperm viability assessment was performed using eosin B (0.5% in saline). A 20 μL sperm suspension sample was placed on a glass slide, mixed with 7 μL eosin B, covered with a slip and examined under a light microscope at a magnification of 400×. Aggregated, dead sperm cells and immotile spermatozoa were excluded from the analysis. Sperm tail morphology was examined by phase-contrast microscopy and classified as follows: (1) with normal flagella, not curved; or (2) with folded flagella. Progressive motility was evaluated when spermatozoa moved actively, either linearly or in a large circle, regardless of speed; all other patterns of motility with absence of progression or immotile spermatozoa were excluded. Analysis for semen parameters was performed counting at least 400 spermatozoa in three independent replicas for each mouse (KO *n* = 11 vs. WT *n* = 10).

### 4.7. Statistics

A two-tailed unpaired *t*-test was used to compare *Kcnj16*^(−/−)^ and *Kcnj16*^(+/+)^ sperm morphology, namely head length, head width, total flagellum length, and midpiece length. Velocities (VSL, VCL, and VAP) were also compared using a two-tailed unpaired *t*-test (GraphPad Prism 6.0 software, San Diego, CA, USA). Alpha was adjusted for multiple comparisons using the Bonferroni correction. To assure reproducibility of findings, a minimum of 6 different testicles per group were used in each experiment. Data are the mean ± standard error of the mean (SEM), with n indicating the number of tests performed in the same experiment.

## Figures and Tables

**Figure 1 ijms-22-05972-f001:**
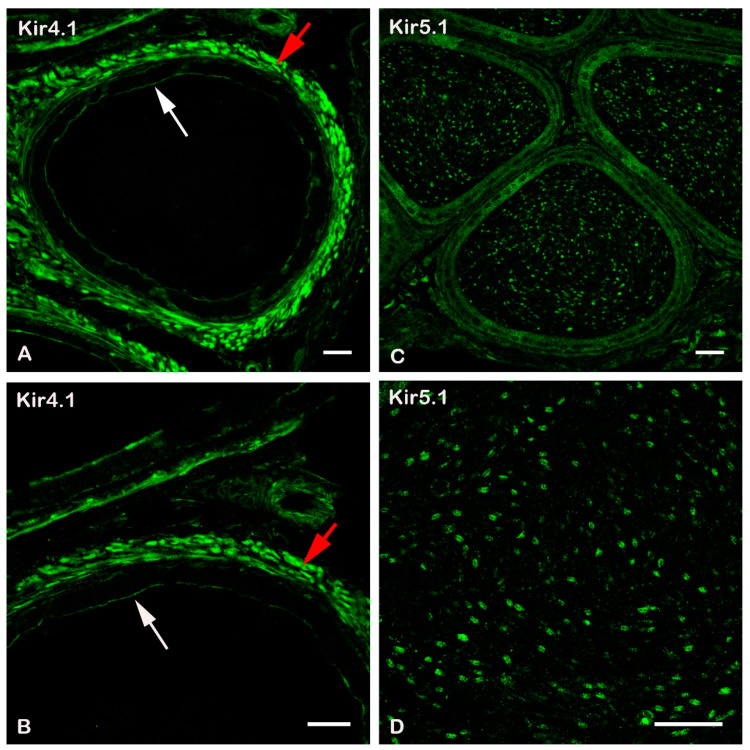
Expression of Kir4.1 and Kir5.1 channels in the cauda epididymis. (**A**) Kir4.1 is expressed in the epithelial cells lining the epididymal ducts (white arrow) and in peritubular smooth muscle of the cauda epididymis (red arrow). Immunoreactivity was absent in the lumen where spermatozoa are located, implying the lack of Kir4.1 expression in these cells. (**B**) Magnified image taken from the upper part of panel A. (**C**) Image showing strong expression of Kir5.1 in spermatozoa. (**D**) Magnification of the lumen of the cauda epididymis. Staining shows the localisation of Kir5.1 subunits in the head of the sperm. Scale bar in (**A**), (**B**), and (**C**) = 25 µm, and in (**D**) = 20 µm.

**Figure 2 ijms-22-05972-f002:**
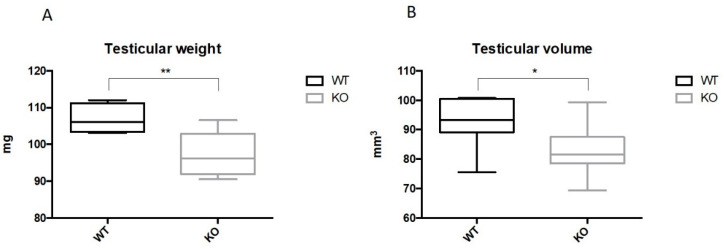
*Kcnj16* (Kir5.1) ablation reduces testicular weight and volume. The box plots report the evaluation of weight (**A**) and volume (**B**) of testis collected from WT and KO mice (*n* = 8; * *p* < 0.05; ** *p* < 0.01).

**Figure 3 ijms-22-05972-f003:**
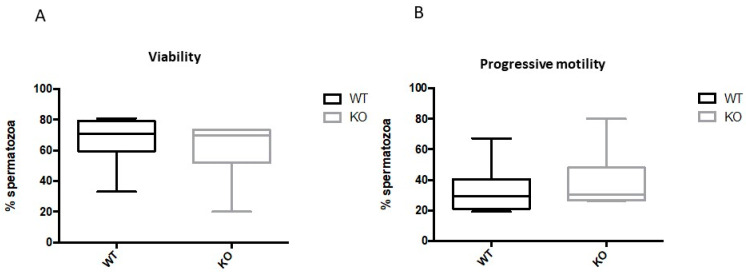
Viability and progressive motility of spermatozoa was comparable in WT and KO mice. The box plots report the evaluation of viability (**A**) and progressive motility (**B**) for spermatozoa collected from WT and KO mice. Both viability and progressive motility were not significantly changed (*p* > 0.05) by KO of Kir5.1 channels.

**Figure 4 ijms-22-05972-f004:**
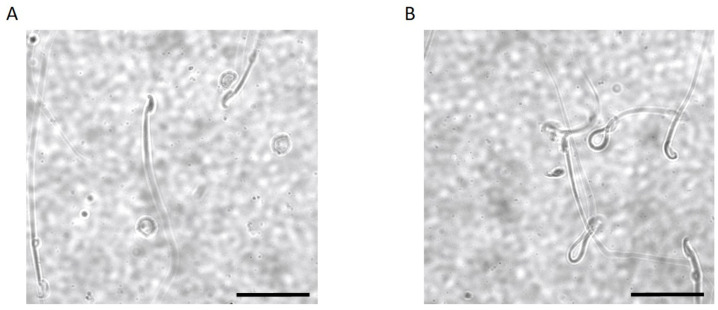
Sperm morphology. WT (**A**) and KO (**B**) mouse sperm observed by phase contrast (Ph2) microscopy. The spermatozoa of both strains of mice displayed normal cell morphology and folded flagella. Folded flagella were more frequent in KO mice. Scale bar in (**A**) and (**B**) = 15 µm.

**Figure 5 ijms-22-05972-f005:**
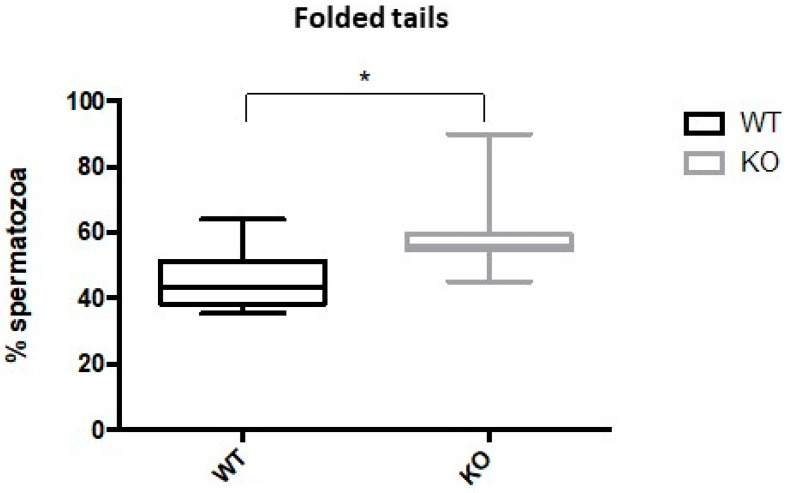
Percentage of spermatozoa with folded tails. Patterns of folded tails from WT and KO mice were analyzed and quantified by phase-contrast microscopy. The box plot showed a significantly higher percentage of spermatozoa with folded tails in the mutant animals compared to WT (* *p* < 0.01).

**Figure 6 ijms-22-05972-f006:**
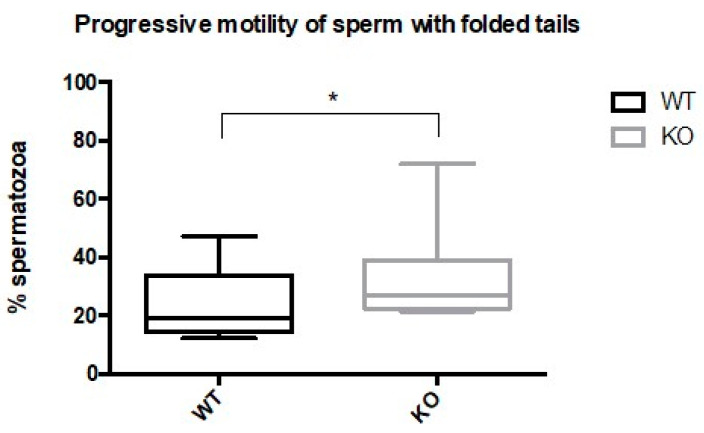
Percentage of folded-tailed spermatozoa with progressive motility was increased in Kir 5.1 KO mice. Progressive motility of folded-tailed spermatozoa from WT and mutant mice was evaluated. The box plot shows the percentage of folded-tailed spermatozoa with progressive motility in WT and KO mice evaluated by bright-field microscopy (* *p* < 0.01).

## Data Availability

Not applicable.

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
