# Peer review of "Kcnj16 (Kir5.1) Gene Ablation Causes Subfertility and Increases the Prevalence of Morphologically Abnormal Spermatozoa"

_ijms, 2021, doi:10.3390/ijms22115972_

Round 1

Reviewer 1 Report

I have no comments. I minor English check should be required. 

Author Response

We are very grateful to Referee n.1, who answered positively to all queries of the questionnaire. The entire paper has been thoroughly revised with significant improvements including English editing. Please, kindly see the changes highlighted in the text.

Reviewer 2 Report

The authors resubmit a previous version of their work.

The revised ms has been improved relatively to the previous one.

Linguistic improvement is needed e.g p. 9 "....testicular underdevelopment..."?. The word hypofertility does not exist in the literature (it is subfertility or infertility)

What other ion channels are neccessary for the functions described? The fact the the specific ablation causes subfertility is a little bit exaggerating because several other channels and membrane receptors are taking part in the processes described. Please elaborate

The Discussion section should be more condensed

Author Response

The entire paper has been thoroughly revised with significant improvements. Please, kindly see the changes highlighted in the text.

Referee n.2 also required “Moderate English Changes” similarly to Referee n.1. Therefore, Prof. Stephen Tucker from the University of Oxford and coauthor of this manuscript performed additional English editing.

Concerning the following sentence in which the referee asked to change “testicular underdevelopment”, linguistic improvements have been performed. In particular, the sentence:”… suggest KCNJ16 as attractive target for genetic screening of patients affected by testicular underdevelopment, morphologically abnormal spermatozoa, altered motility and subfertility” has been changed as follows: “……suggest KCNJ16 as attractive target for genetic screening of patients that exhibit abnormal testicular development, altered sperm morphology and subfertility.”

The word “hypofertility” was substituted with the word “subfertility” in the entire manuscript.

The referee asks: “What other ion channels are necessary for the functions described? The fact the specific ablation causes subfertility is a little bit exaggerating because several other channels and membrane receptors are taking part in the processes described. Please elaborate.”

The referee also asks that “the discussion section should be more condensed”.

In order to address the criticism briefly and keep the discussion section more condensed we wrote:  “This overt phenotype of Kir5.1 KO mice suggests that Kir5.1 channels are not dispensable and cannot be compensated by other members of the Kir family or any other K+ channel and may be related to their extreme sensitivity to intracellular pH. Cytoplasmic acidifications strongly inhibits these channels which will depolarize the membrane, whereas, alkalization will hyperpolarize the membrane potential.”

Several sentences in the discussion were deleted to shorten this section.

This manuscript is a resubmission of an earlier submission. The following is a list of the peer review reports and author responses from that submission.

Round 1

Reviewer 1 Report

Dear Authors,

I have read the manuscript entitled “Kcnj16 (Kir5.1) gene ablation increases the prevalence of morphologically abnormal spermatozoa”, which is aimed to investigate the role of the Kir5.1 protein in sperm morphology and motility, by using a Kcnj16 knock out mice model. The Authors reported a significant increase in the number of spermatozoa with twisted flagella and of spermatozoa with moving twisted shape. Therefore, they speculate a role for this protein in fertilization and fertility. The manuscript is well-written, the English is fluent, the methods fit with the study purpose.

Major comments

1) I would suggest showing results as a percentage value, more than as an absolute value (number of spermatozoa). Accordingly, sperm morphology and motility are usually presented as a percentage value (e.g. among 200 or 400 observed spermatozoa), and the WHO manual recommends the same for human sperm analysis.

2) Are the sperm number (concentration and total count) and the testicular volume different in KO vs. WT mice?

3) Did the Authors assessed whether any difference occurred in the semen pH among KO vs. WT mice?

4) Would it be possible to add electrophysiological experiments to demonstrate that a different pHi (or membrane potential) or different extracellular K levels occur in spermatozoa or in the semen (respectively) of KO vs. WT mice?

Minor comments

1) Lines 59 and 62: “Casamasssima”. Please correct.

2) Line 141: please rephrase.

Author Response

 Major comments

1) I would suggest showing results as a percentage value, more than as an absolute value (number of spermatozoa). Accordingly, sperm morphology and motility are usually presented as a percentage value (e.g. among 200 or 400 observed spermatozoa), and the WHO manual recommends the same for human sperm analysis.

We thank the reviewer very much for these comments. We are now showing results as a percentage value, according to the Reviewer’s valuable suggestion.

2) Are the sperm number (concentration and total count) and the testicular volume different in KO vs. WT mice?

We are very grateful to the reviewer for this question, as by performing the experiment that s/he suggested we uncovered that the genetic inactivation of Kcnj16 also resulted in smaller testis. Indeed, “the weight and volumes of testis collected from 2-months-old KO mice were significantly smaller than the WT (Fig2. A, B). However, the body weights of WT and mutant mice, from which the testis were collected, were not statistically different (26.8±0.8g vs 26.0±1.7g; n=8).”

As for the concerns on the count of spermatozoa, we showed that the number of those displaying progressive motility was slightly higher in Kir5.1 KO (166 ± 52) compared to WT mice (132 ± 43), although the computed values were not statistically significant (Fig. 3B).

3) Did the Authors assessed whether any difference occurred in the semen pH among KO vs. WT mice?

We have performed the useful experiment appreciably suggested by the referee and reported that: “When the luminal contents of cauda epididymidal fluid were immediately spread onto pH strips without delay in the time during collection and measurement, pH values obtained for the mutant and wild-type mice were not different.” 

Would it be possible to add electrophysiological experiments to demonstrate that a different pHi (or membrane potential) or different extracellular K levels occur in spermatozoa or in the semen (respectively) of KO vs. WT mice?

We thank the referee for the fantastic experiments that s/he suggested. However, patch-clamp recordings from spermatozoa are very difficult to perform and only very few labs in the world have developed such a crucial methodology. One important aim of our paper is indeed to draw the attention of the investigators that currently perform such experiment to demonstrate that different pHi or different extracellular K levels change spermatozoa membrane potential through Kir5.1 channels and consider the possibility to characterize the hypothesized existence of a crosstalk between Hv1, Kir5.1 and CatSper channels. Unfortunately, our lab is unable to perform such experiments.

Minor comments

1) Lines 59 and 62: “Casamasssima”. Please correct.

2) Line 141: please rephrase.

We have corrected the typos and rephrased.

Reviewer 2 Report

The authors tried to find any correlation between a specific deletion of one gene that participates in sperm movement and sperm morphology.

They found that the specific deletion has an effect on sperm tail morphology but it does not affect viability, fertility and embryogenesis.

According to  the resutls it seems that this specific deletion is a silent mutation that does not have any effect overall

The authors should state the acceptable percent of bad morphology and to compare it with the deleted one. I mean that if the deletion decreases the morphology of spermatozoa to acceptable levels (although significantly decreased) of morphology and if the deletion produces spermatozoa that remain vital, fertile, for what reason is significant this deletion/ablation?

Author Response

The authors tried to find any correlation between a specific deletion of one gene that participates in sperm movement and sperm morphology.

They found that the specific deletion has an effect on sperm tail morphology but it does not affect viability, fertility and embryogenesis.

According to the results it seems that this specific deletion is a silent mutation that does not have any effect overall.

The authors should state the acceptable percent of bad morphology and to compare it with the deleted one. I mean that if the deletion decreases the morphology of spermatozoa to acceptable levels (although significantly decreased) of morphology and if the deletion produces spermatozoa that remain vital, fertile, for what reason is significant this deletion/ablation?

We thank the reviewer very much for these comments. This concern was also raised by referee #1 and we are now showing the percentage of bad morphology sperm and compared it with the deleted one. Furthermore, we fully agree and are very grateful to the reviewer for posing the important question: “ for what reason is significant this deletion/ablation?”. We were able to retrieve the relevant data, from the registry of our laboratory and that from the animal house, concerning the breeding performance of Kcnj16 knock-out (KO) mice during a 2-year-long period. Overall, we observed that 20% of mutant male mice were infertile. Furthermore, 50% of animals older than 3 months were unable to breed. By contrast, 100% of wild-type (WT) mice were fertile and became infertile at over 24 months of age, consistent with previous findings (Franks and Payne 1970).  Furthermore, the genetic inactivation of Kcnj16 also resulted in smaller testis. Indeed, the weight and volumes of testis collected from 2-months-old KO mice were significantly smaller than the WT (Fig2. A, B). However, the body weights of WT and mutant mice, from which the testis were collected, were not statistically different (26.8±0.8g vs 26.0±1.7g; n=8). All these new results are now included in the manuscript. Thus, we propose that: “ablation of Kcnj16 gene identifies Kir5.1 channel as an important element contributing to testicular development, male hypofertility, decline in fertility with increasing age, spermatozoan flagellar morphology and motility.”

Round 2

Reviewer 1 Report

Congratulation for this very nice study!

I have no futher comments.